# Experimental Study on the Solidification of Uranium Tailings and Uranium Removal Based on MICP

**Lin Hu** [1,2], **Zhijun Zhang** [1,2], **Lingling Wu** [1,2,*] , **Qing Yu** [1,2], **Huaimiao Zheng** [3], **Yakun Tian** [1,2] and **Guicheng He** [1,2]

---

1 School of Resource & Environment and Safety Engineering, University of South China, Hengyang 421001, China; 343737982229@usc.edu.cn (L.H.); 130000148665@usc.edu.cn (Z.Z.); 2009000547@usc.edu.cn (Q.Y.); 2017000012@usc.edu.cn (Y.T.); 2005000509@usc.edu.cn (G.H.)
2 Hunan Province Engineering Technology Research Center for Disaster Prediction and Control on Mining Geotechnical Engineering, Hengyang 421001, China
3 School of Economics, Management and Law, University of South China, Hengyang 421001, China; 2004001109@usc.edu.cn
* Correspondence: wllshmily@foxmail.com

**Abstract:** The governance of uranium tailings aims to improve stability and reduce radionuclide uranium release. In order to achieve this goal, the uranium removal solution test and uranium tailings grouting test were successively carried out using microbially induced calcium carbonate precipitation (MICP) technology. The effect of MICP on the reinforcement of uranium tailings and the synchronous control of radionuclide uranium in the tailings were discussed. The solution test results show that *Sporosarcina pasteurii* could grow and reproduce rapidly in an acidic medium with an initial pH of 5. The uranium concentration decreased with the increase in MICP reaction time, and the removal efficiency reached 60.9% at 24 h. In the solidification test of tailings, the strength of tailings improved significantly after 12 days of reinforcement, with an increase in the cohesion of tailings by 2.937 times and an increased internal friction angle of 8.393°. The peak stress value of solidified tailings at the surrounding pressure of 50 kPa increased by 1.87 times, and the uranium concentration in the discharge fluid decreased by 76.91% compared to the blank group. This study provides valuable insights and references for safely disposing of uranium tailings.

**Keywords:** uranium tailings; solidification; uranium removal; microorganism; calcium carbonate





## 1. Introduction

In recent years, there has been a surge of interest in microbially induced calcite precipitation (MICP) as a highly effective and sustainable reinforcement technology, as evidenced by numerous domestic and international studies [1]. These studies demonstrate that MICP can induce the formation of crystals with exceptional cementing properties through the metabolic activities of microorganisms. When applied to geotechnical materials, this technology significantly boosted their strength [2,3]. MICP technology empowers an ecologically friendly approach to enhancement since it can improve the stiffness, bearing capacity, permeability, and liquefaction resistance of earth and rock. Song et al. [4] conducted several cycles of MICP via grouting on sandstone. This achieved remarkable results: a 229% increase in uniaxial compressive strength, a 179% increase in elastic modulus, and a 177% increase in brittleness index compared to the pre-grouting conditions. The overall mechanical properties and permeability of sandstone were shown to be primarily impacted by the amount of cemented minerals present, which, in turn, directly govern the microscopic distribution of $CaCO_3$, amplifying the efficacy of bio-cementation in sandstone. Banik et al. [5] explored the microstructures and mechanical characteristics standard sand specimens of after MICP reinforcement. The resulting stress–strain data and the strength gain observed under different pore volumes of cementing fluids provided insights into

quantifying the strength of microbially reinforced sand and enhancing the low-strain shear modulus. Zhao et al. [6] carried out a solution and cyclic triaxial tests while regulating the concentration of NaCl and observed that the stiffness and cyclic resistance of the additive solids gradually decreased as the NaCl concentration increased. However, these values remained higher than those of unreinforced sand, and it was discovered that the decline in liquefaction resistance was caused by the conversion of calcium carbonate crystals from clusters to single crystals induced by the technology. Xiao et al. [7] leveraged the application of MICP-reinforced coral sand by acclimating microorganisms to various environments to improve their enzymatic activities, developing a novel three-stage reinforcement method. Experimental results confirm that this method considerably increased the strength of MICP treatment and brought about a 32% increase in induced calcium carbonate content. Additionally, suitable cementing solution concentrations proved advantageous for enhancing reinforcement strength.

In addition to studying the reinforcement effect of MICP on soils, considerable attention has been given to the application of MICP technology for the remediation of heavy metal pollution. Chen et al. [8] used bacillus *pasteurii* to immobilize lead, reducing the leaching of $Pb^{2+}$ in water by 76.34%. Achal et al. [9] used a copper-resistant strain of *Kocuria* flava CR1 isolated from a mining area. The bacteria showed a high urease activity under culture medium conditions at an initial $Cu^{2+}$ concentration of 1000 mg/L, pH value of 8.0, calcium source of $CaCl_2$, urea concentration of 2%, and a temperature of 30 °C. At that time, the $Cu^{2+}$ removal rate was 97%. Li et al. [10] discovered a strain of *Sporosarcina pasteurii* UR31 in soil that is highly tolerant to zinc. When $ZnCl_2$ was introduced to the culture medium at an initial concentration of 2 g/L, this strain was able to remove as much as 99% of the zinc. Jalilvand et al. [11] isolated two rhizobacterial strains, *Stenotrophomonas rhizophila* A323 and *Variovorax boronicumulans* C113, from grassroots. The study found that these strains exhibited cadmium tolerance and produced urease. In aqueous solutions, these bacteria were discovered to transform more than 70% of soluble Cd into insoluble carbonate mineral forms. Moreover, the researchers utilized isolated strains of *Sporosarcina pasteurii* to eliminate $Cd^{2+}$ and accomplished a removal rate of 97.15%. These studies show that MICP technology holds promising potential and has good prospects for the bioremediation of heavy metal pollution.

Uranium tailings contain a large amount of low-grade uranium ore, and the uranium in the tailings accounts for about 1~5% of the original ore. It contains 85% of the radioactivity found in raw ore [12]. Although uranium tailings have a low average grade of uranium, ranging from 0.005% to 0.03%, the total amount of uranium present must be taken into consideration and addressed. Additionally, the long-term storage of uranium tailings causes severe radioactive pollution to the surrounding environment and results in a significant waste of uranium resources [13,14]. Therefore, the current governance objective of uranium tailings (slag) ponds is to address the above environmental issues at the source by implementing reasonable and practical measures to enhance their stability and reduce the release of residual radionuclides [15].

The objective of this study is to enhance the stability of uranium tailings and prevent radioactive contamination. Through the utilization of MICP technology, the experimental microbial grouting reinforcement of uranium tailing sand was conducted to investigate the impact of MICP on uranium tailings reinforcement and radionuclide control. As opposed to the single focus on the reinforcement effect in most current studies, the synchronous control effect of pollutants is also considered in this study. Microbial grouting reinforcement is characterized by a low cost and short reinforcement time and is environmentally friendly. Compared with the inclusions of tailings formed via traditional chemical grouting, the biobased body is more convenient for grinding dissociation and recovery, which is conducive to the recovery and comprehensive utilization of effective resources in cemented tailings. This is important for the theoretical innovation and technical promotion of the safe disposal of uranium tailings.

## 2. Materials and Methods

### 2.1. MICP Aqueous Solution Tests

#### 2.1.1. Purposes

To examine the effectiveness of mineralizing bacteria in sand grouting tests on uranium tailings, the activity and ability of the mineralizing bacteria to perform normal mineralization reactions were determined in an acidic and radioactive solution.

#### 2.1.2. Materials

The test was carried out using *Sporosarcina pasteurii* as the dominant species, purchased from the American Type Culture Collection (Rockefeller, MD, USA), and numbered ATCC11859. The liquid medium consisted of 15 g/L casein peptone, 5 g/L soy peptone, 5 g/L NaCl, 20 g/L urea, and 1000 g/L distilled water. The solid medium was added to 20 g/L agar based on the liquid medium. The cement solution comprised 1 M urea solution and 1 M $CaCl_2$ solution with a volume ratio of 1:1.

#### 2.1.3. Methods

In order for the subsequent mineralization reaction to be effective in uranium tailings, the acidity of the solution and the concentration of uranium must be equal to or stronger than that of uranium tailings. Therefore, in order to set the appropriate initial pH value and uranium concentration for the solution, the pH value and uranium concentration of the uranium tailings must be measured first. Uranium tailings samples from a local large uranium tailings pond in Hunan Province were soaked in deionized water for 48 h, the soaking solution was sampled, and a method was used to determine pH and uranium concentration [16]. The result was a pH of 5.3 and uranium concentration of 0.365 mg/L. With reference to this value, the initial pH of the solution was set to 5 and the initial concentration of uranium was set to 2 mg/L.

After the mineralization reaction was completed, the uranium concentration was determined using inductively coupled plasma—mass spectrometry (ICP-MS). When the concentration of uranium was measured using ICP-MS, the quality control methods were as follows: ① Because the pH of the solution after the mineralization reaction increases and becomes alkaline, nitric acid was used to dissolve the filtered solution. ② Before testing the sample, the machine was preheated for 30 min, so that the temperature of the instrument itself was stable, and the wavelength was unaffected by the temperature. ③ The calibration was first carried out with the standard liquid of uranium. ④ The argon gas flow rate was slowly adjusted to 15 L /min, and the auxiliary gas flow rate was 2 L /min.

When the concentration of the original bacterial solution reached its maximum, an appropriate amount of the bacterial solution was transferred to the culture above the medium for expansion. After expansion, 100 mL of the bacterial solution was poured into the reaction vessel. Simultaneously, 400 mL of cement solution (concentration specified in Section 2.1.2) was slowly injected through a peristaltic pump. Then, the reaction vessel was placed on a magnetic stirrer and stirred for 24 h. During the experiment, every 2 h, the bacterial solution was taken out to determine the pH, OD600 value, and uranium concentration, with a total of 12 determinations. After 24 h, the precipitate was taken for further testing.

The main detection instruments involved in this study are shown in Table 1.

**Table 1.** List of testing instruments.

| Experimental Instrument | Company | Model |
|---|---|---|
| pH | Mettler Toledo, Shanghai, China | S220-K |
| OD600 | Eppendorf, Hamburg, Germany | Biophotometer |
| ICP-MS | Agilent, Santa Clara, CA, USA | 7700× |
| XRD | Hitachi, Tokyo, Japan | Regulus8100 |
| SEM | Bruker, Karlsruhe, Germany | Bruker D8 ADVANCE |
| CT | Zeiss Xradia, Turingen, Germany | 410Versa |
| Triaxial testing | Teco, Nanjing, China | TKA-TTS-IS |

### 2.2. Grouting Tests on Uranium Tailings

2.2.1. Purposes

In order to evaluate the effectiveness of MICP technology in stabilizing uranium tailings and removing uranium, experiments were conducted on the uranium tailings using MICP technology, as per the findings presented in Section 2.1.

2.2.2. Materials

The experiment tailings were collected from a large uranium tailings pond in Hunan Province, China. The size of the uranium tailing particles is relatively uniform, mainly being translucent granular minerals with metallic luster and little debris. Impurities such as quartz can be seen. The hardness of the tailings can be felt by kneading them by hand. The relative density of the uranium tailings is 2.53, with a natural dry density of $1.446\ \text{g/cm}^3$ and a void ratio of 58.8%. In order to determine the primary parameters of particle size for uranium tailings, a series of particle analysis tests were conducted on the tailing sand. Based on the test results, the cumulative curve of particle size distribution of tailings is drawn, as shown in Figure 1. According to the curve, the relevant parameters of tailings particle size were determined, such as the restricted particle size $d_{60} = 1.00$ mm, effective particle size $d_{10} = 0.16$ mm, median particle size $d_{30} = 0.5$ mm, inhomogeneity coefficient $C_U = 6.25 > 5$, and curvature coefficient $C_C = 0.15625 > 1$. According to the Standard for Engineering Classification of Soil, the size distribution of uranium tailings is continuous and particle size distribution is good [17].

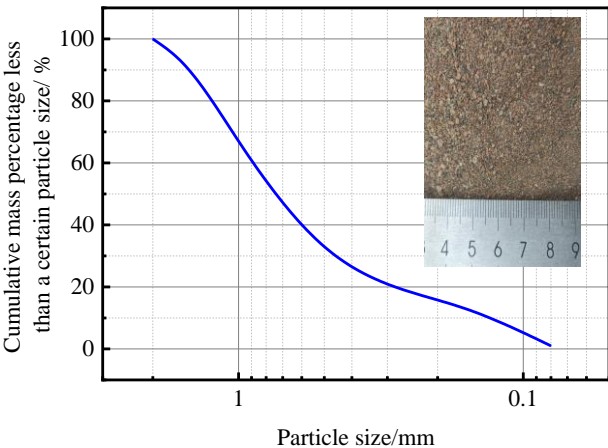

**Figure 1.** Cumulative curve of particle gradation of tailings.

The microorganisms, culture medium, and cement solution used for grouting are the same as those described in Section 2.2.1.

2.2.3. Test Devices and Methods

The tailing sand was filled and compacted into the grouting reinforcement device before the test. The grouting reinforcement device is shown in Figure 2. The thickness, width, and length of tailings are 30 cm, 40 cm, and 100 cm, respectively. The slope of tailings is 30°, and the volume of tailings is calculated as follows: 100 cm × 40 cm × 30 cm − 1/2 × 40 cm × 30 cm × 60 cm × cos30° = 88,800 cm³. In tailings with a moisture content of approximately 10% and void ratio of 58.8%, the total amount of grout required was calculated as 88.8 × 48.8 = 43.33 L [18]. To make the calculation easier, the total grouting volume was fixed at 44 L, which is composed of 8.8 L of bacterial liquid, 17.6 L of $CaCl_2$ solution, and 17.6 L of urea solution. The reaction liquid was injected in a specific order, starting with bacterial solution, followed by urea solution, and $CaCl_2$ solution. The rate of grout injection was 20 mL/min, and the grouting was carried out once per day. After each grouting, the drain pipe valve was opened to collect residual liquid. This process was repeated for 12 days, and, at the end, samples were taken for testing and analysis.

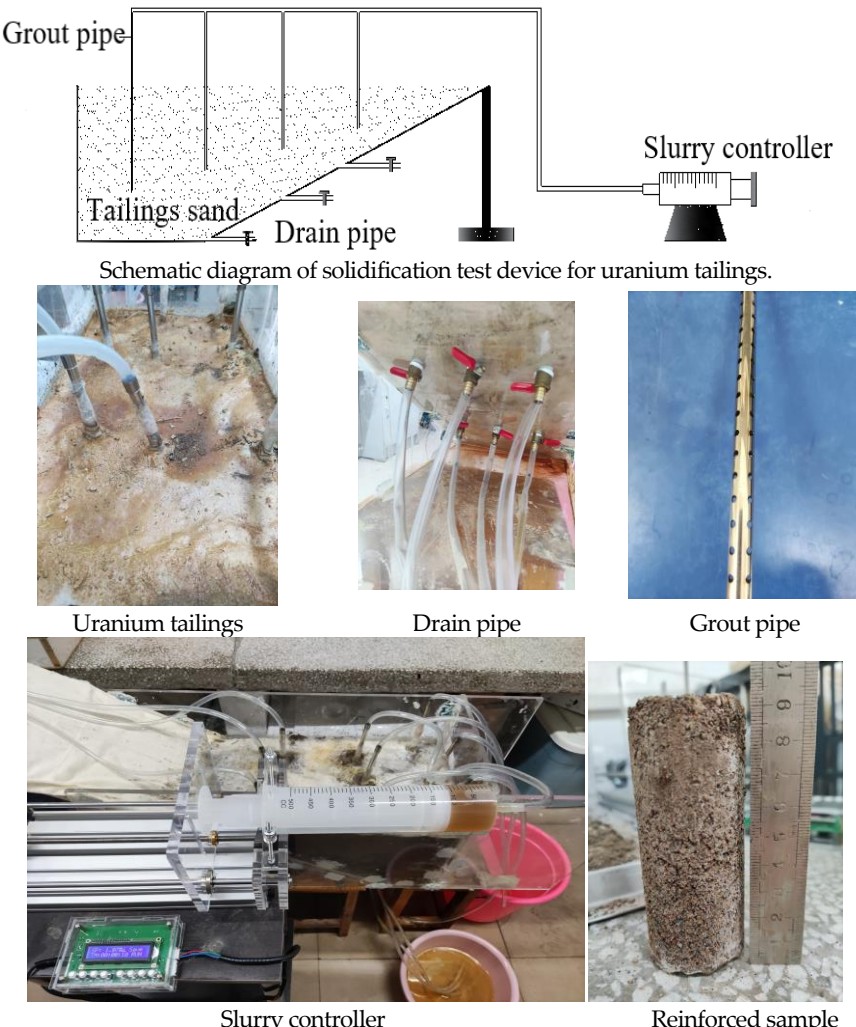

Figure 2 images labeled:
Schematic diagram of solidification test device for uranium tailings.
Uranium tailings    Drain pipe    Grout pipe
Slurry controller    Reinforced sample

**Figure 2.** Tailing reinforcement device diagram.

## 3. Results and Analysis

### 3.1. Solution Test Results

3.1.1. Biochemical Characterization of Bacteria

The change in pH significantly impacts the physiological metabolic activities and nutrient absorption abilities of *Sporosarcina pasteurii*. Moreover, pH has a more significant impact on intracellular and extracellular urease activity produced during bacterial metabolism. Therefore, the biochemical characteristics of the bacteria are reflected by measuring the pH and the bacterial cell density OD600 value of the bacterial liquid. Every 2 h, the pH of the bacterial solution was measured, while the OD600 value was measured using a protein–nucleic acid analyzer. According to the test results, the growth and pH change curves of the bacteria were drawn and are shown in Figure 3.

Figure 3 shows that when the initial pH is 5 and the uranium mass concentration is 2 mg/L, the number of OD600 bacteria linearly increases within 4–8 h, indicating vigorous multiplication. However, with the gradual depletion of nutrients and accumulation of metabolic waste, the growth rate of OD600 bacteria decrease between 10 and 24 h. It can also be seen from Figure 3 that the pH value rapidly increases and stabilizes, eventually staying at approximately 10, Indicating that the bacteria can multiply in the medium with an initial pH of 5. The pH increased almost linearly within 0~6 h, and then stabilized from 6 to 8 h. The pH values showed the same trends as the OD600 values. The pH values varied within 0~6 h, mainly due to the rapid increase in the number of bacteria and urea hydrolysis rate caused by urease, which produced a considerable number of carbonate and ammonium

ions. When the pH value reached around 10 at 6 h, it provided an alkaline environment for bacterial growth and reproduction. However, the ability of bacteria to multiply reached its maximum, and the nutrient resources in the culture medium were consumed at the later stages, resulting in the pH value gradually stabilizing and slowly decreasing.

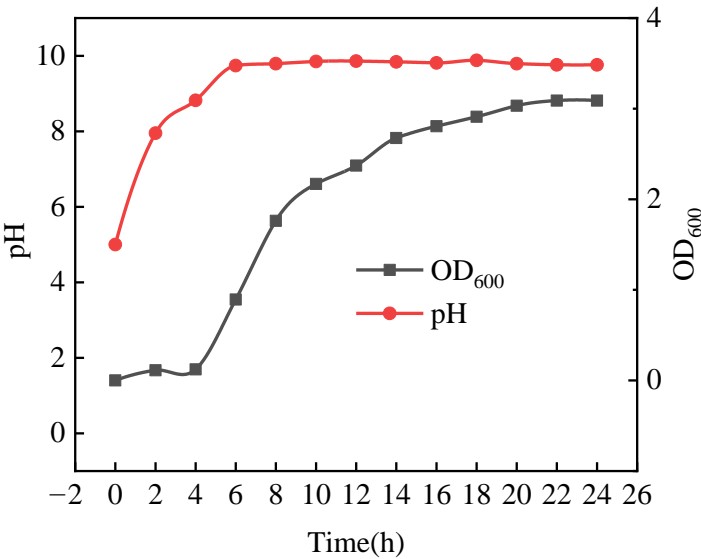

**Figure 3.** Growth curve and pH change curve of *Sporosarcina pasteurii*.

### 3.1.2. Changes in Uranium Concentration

The uranium concentration in the solution was measured every 6 h during the reaction, and the results are shown in Figure 4. According to Figure 4, the uranium concentration significantly decreases within the first 10 h and slightly increases after 12 h. This is because some uranium was adsorbed in the form of unstable combinations of anions and cations, resulting in a small amount of solidified uranium being reintroduced into the solution over time. When the reaction time was 24 h, the rate of uranium removal was 60.9%.

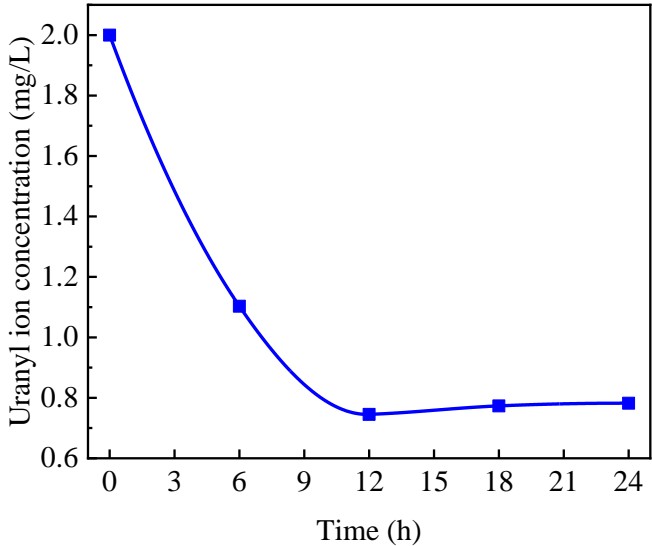

**Figure 4.** Change curve of uranium concentration in solution.

### 3.1.3. Mineralization Product Test Results
X-ray Diffraction (XRD)

After the reaction, the resulting crystal precipitate was washed, dried, and weighed, yielding a mass of 4.226 g. The crystalline products appeared slightly yellow and white

under visual observation, with stacked and arranged crystals, large particle agglomeration and interparticle stacking connections, dense crystal–crystal stacking, rich pores, a uniform crystal texture, and good structure. The mineralized products were taken for XRD test analysis, and the results are shown in Figure 5. Figure 5 shows that the crystalline product is calcium carbonate crystals, which contain crystalline forms of calcite and vaterite.

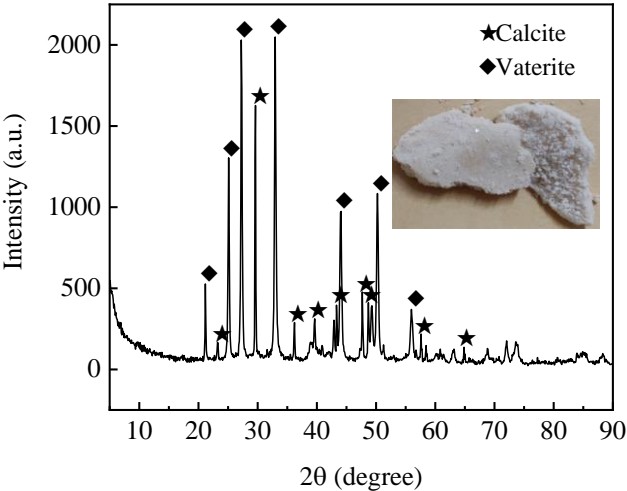

**Figure 5.** XRD test results of mineralized products.

Scanning Electron Microscopy (SEM)

SEM tests were carried out on calcium-carbonate-mineralized samples, and the results are shown in Figure 6. The scanned images from Figure 6 indicate that the main forms of calcium carbonate crystals are regular and uniformly shaped vaterite and calcite with various shapes (massive and granular) and uneven grain sizes, consistent with XRD results. The scanned images also show that the crystal surface is uneven and densely porous, and vaterite and calcite overlap with each other and stack in clusters, creating a structure conducive to stabilizing the crystals.

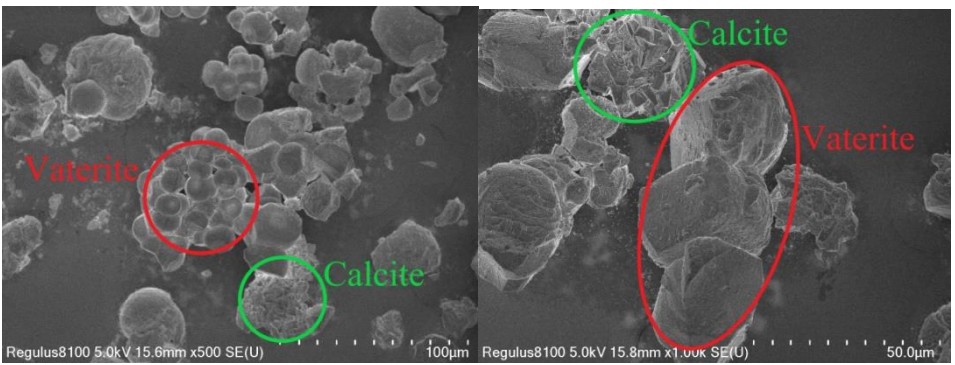

**Figure 6.** SEM image of calcium carbonate crystal.

### 3.2. Test Results Uranium Tailing Sand Grouting

3.2.1. The Triaxial Test Results

The strain-controlled triaxial apparatus based on the Mohr–Coulomb strength theory (as shown in Formula (1)) was utilized for the undrained shear test. Confining pressures of 50 kPa, 100 kPa, and 200 kPa were set, and the axial force was applied immediately after the specimen became stable, resulting in the tailings specimen's shear failure under undrained conditions. During the test, critical parameters, such as the shear strength index of tailings, were monitored and analyzed, and their cohesion and internal friction angle

were calculated. The stress–strain curves of tailing samples strengthened by each group of tailings under the consolidated undrained triaxial shear test are depicted in Figure 7.

$$\tau = \sigma \cdot \tan \varphi + c \tag{1}$$

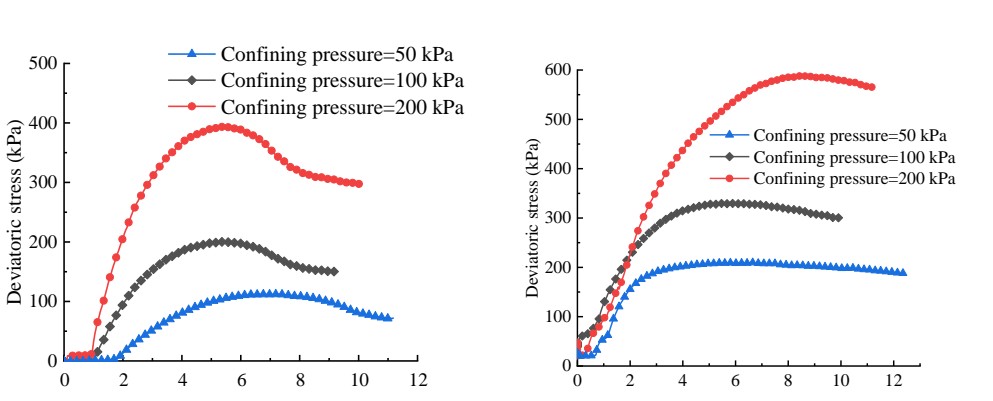

**Figure 7.** Stress–strain relation curve of tailing samples.

$\tau$ is shear strength; $\sigma$ is normal stress; $\varphi$ is the angle of internal friction; c is the cohesion.

The stress–strain curve reveals the changing trend in the stress–strain responses of the tailings samples under axial load, broadly categorized into four stages. In the initial loading stage, the pores within the tailing samples begin to compact under axial force, leading to small transverse deformations and a decrease in sample volume over time. This is the pore compaction stage. Moving into the second stage, the load increases and the deviator stress of the sample reaches its peak value, followed by entry into the third stage. In this stage, the deviator stress gradually decreases, and the specimen incurs some damage. Continued loading leads to shear failure along cracks within the sample. Entering the fourth stage, axial deformation increases while deviator stress decreases and gradually stabilizes. Upon reaching the set axial strain value, the specimen experiences damage, although it still displays a certain degree of strength due to mutual embedding and occlusion between particles.

Compared to the unstrengthened group, the deviatoric stress of the MICP grouting group was enhanced. The peak value increased by 1.87 times at a confining pressure of 50 kPa, the peak value increased by 1.65 times at a confining pressure of 100 kPa, and the peak value increased by 1.49 times at a confining pressure of 200 kPa.

The Mohr–Coulomb strength theory formula was used to process triaxial shear strength test data, obtaining the characteristic values of mechanical parameters of tailings samples under different confining pressures, as displayed in Table 2. It can be seen from Table 2 that under the same test conditions, the cohesion of unreinforced uranium tailings is 5.564 kPa and the angle of internal friction is 17.891°. By contrast, the cohesion of the reinforced tailing samples greatly improved. The cohesion of the tailing samples after 12 days of reinforcement increased by 2.937 times compared to the original reinforcement. The internal friction angle increased by 8.393°. This shows that after 12 days of cyclic reinforcement, the reinforcement of tailings had a certain effect.

**Table 2.** Mechanical characteristic parameters of the triaxial test.

| Test Group | Confining Pressure (kPa) | Deviatoric Stress (kPa) | Cohesion (kPa) | Internal Friction Angle (°) |
|---|---|---|---|---|
| original tail sand | 50<br>100<br>200 | 112.15928<br>200.29918<br>393.86521 | 5.564 | 17.891 |
| Reinforcement (12 days) | 50<br>100<br>200 | 199.62468<br>329.99297<br>588.21775 | 21.906 | 26.284 |

### 3.2.2. Comparative Analysis of the Micro-Morphology of Cement

To analyze the microstructure of samples of original tailings and reinforced tailings, a scanning electron microscope (SEM) was employed. From Figure 8, it can be seen that the original tailings have relatively flat and smooth surfaces, without obvious pores and cracks, and there are few surface attachments.

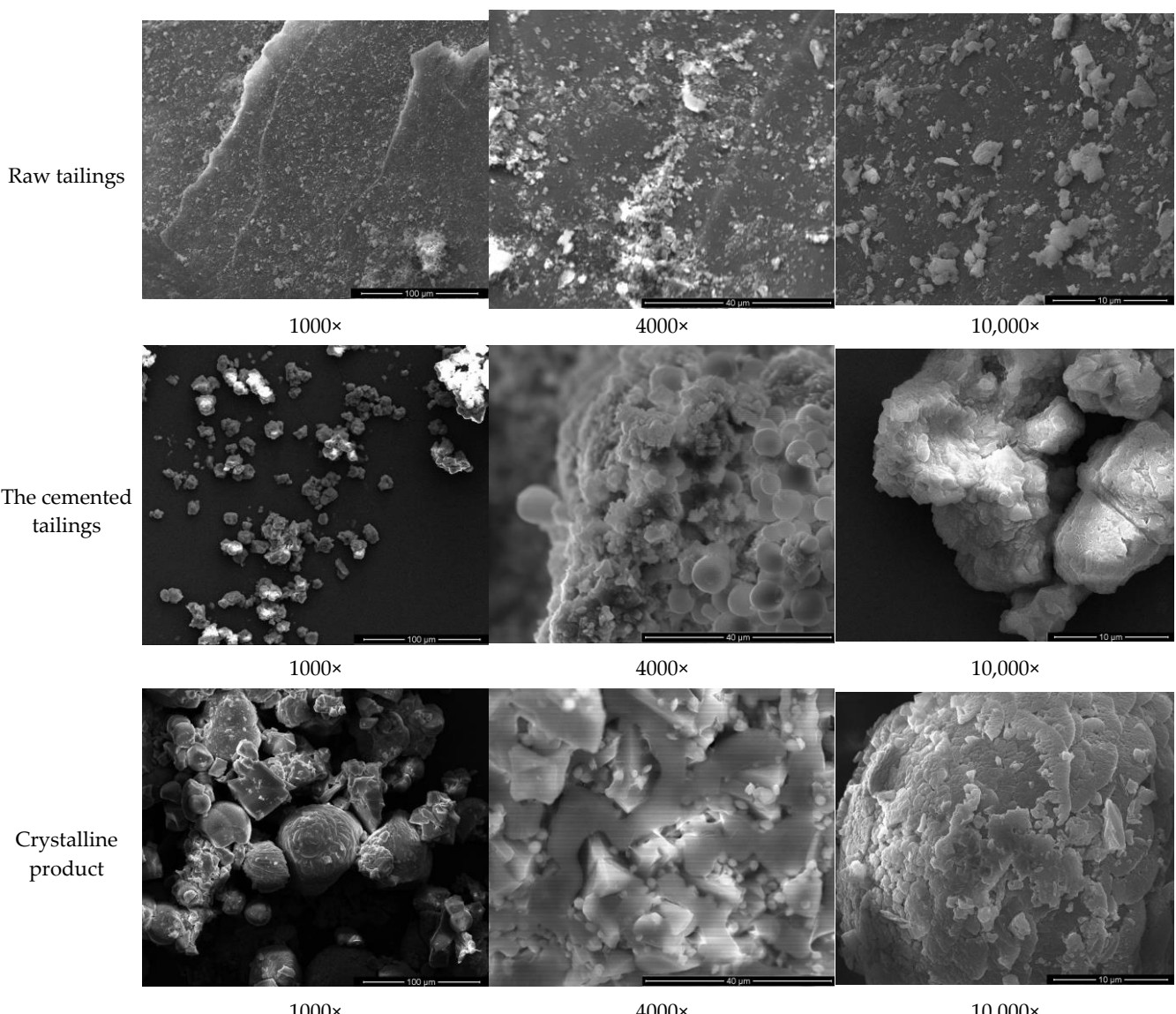

**Figure 8.** SEM images of the tailings before and after reinforcement.

The SEM image of the cemented sand shows that after cementation, many calcium carbonate crystals adhere to the surface of the tailings, and the crystals are stacked and arranged. Particle aggregation is clearly visible, uneven, and porous. From the $4000\times$ image, it can be seen that calcium carbonate almost completely covers the tailing particles. This coverage and cementation form an intricate stacking pattern that is irregular in connection and stacking, which is beneficial for improving compactness and enhancing the consolidation effect.

It is clear from the SEM image of calcium carbonate crystal that the generated crystal is mainly composed of vaterite with regular and uniform shape and calcite with various shapes (massive and granular) and uneven grain size. Vaterite and calcite are embedded

and distributed, stacked in clusters, and bonded together. The image results are consistent with the XRD detection results of mineralized products in the solution experiment.

### 3.2.3. XRD Detection

Since the SEM test only provides the surface morphology information of calcium carbonate crystals, it cannot be used as a sole basis for determining the crystal form. Therefore, an XRD test was conducted to obtain crystal parameters and characterize the substance structure. The representative samples were selected for XRD detection, and the results are shown in Figure 9. From Figure 9, it can be seen that the composition of the tailing sand particles after cementation is mainly calcite, vaterite, and quartz. The calcium carbonate crystal form mainly consists of calcite and vaterite, which is consistent with the SEM test results.

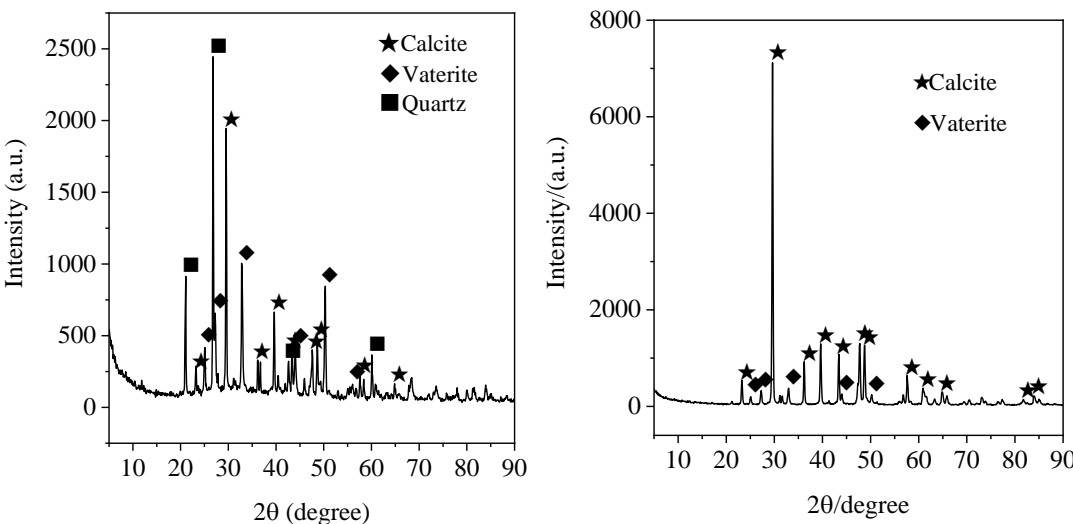

**Figure 9.** XRD detection diagram of tailings.

### 3.2.4. Quantitative Characterization of Pore Structure Based on CT Scanning

After being treated with microbial grouting, the uranium tailing samples were scanned using computed tomography (CT) technology. Three-dimensional imaging was utilized to extract information on porosity and particle characteristics, allowing for the analysis of microscopic processes and quantitative characterization of structural changes during the grouting process [19]. The scanned sample data were reconstructed in three dimensions using visualization software to obtain the 3D pore image of the tailing column, as shown in Figure 10. From Figure 10, it can be observed that the pore size and number of tailing samples after 8 days of grouting are significantly greater than those after 12 days of reinforcement. It can also be seen from the figure that the pore volume in the upper part of the sand column is smaller than that in the lower part.

To further quantify the change in porosity inside the tailing column, the porosity of each layer along the Z-axis direction (from bottom to top) of the column at different periods during the grouting process was statistically analyzed using Dragonfly software (Dragonfly 2021.1 Build 977), and the results are shown in Figure 11.

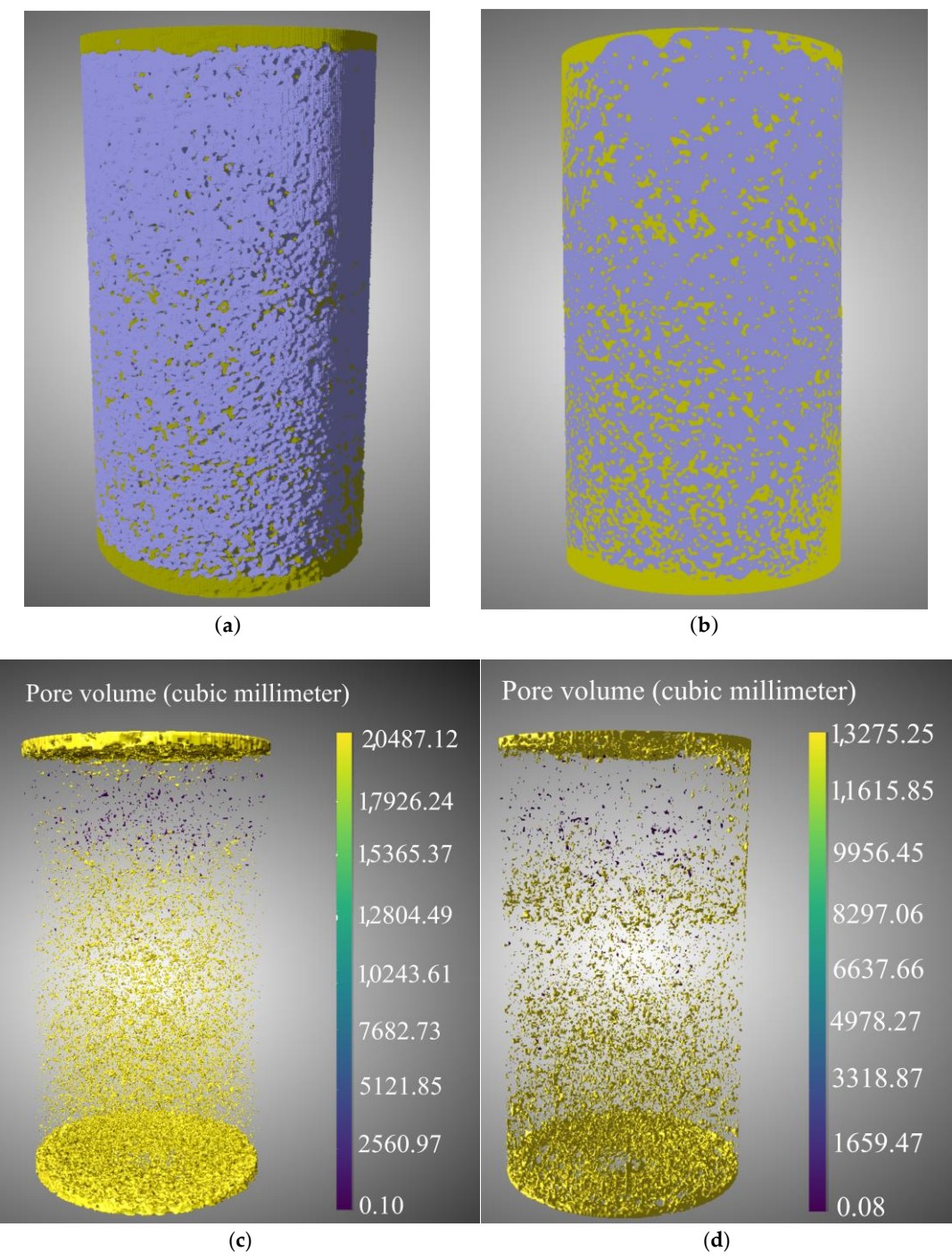

**Figure 10.** Three-dimensional pore structure of tailing samples. (**a**) Reinforced 8-day 3D re-composition of the tailing column. (**b**) Reinforced 12-day 3D re-composition of the tailing column. (**c**) Reinforced pore volume distribution of the tailing column for 8 days. (**d**) Reinforced pore volume distribution of the tailings column for 12 days. Description: In subfigures (**a**,**b**), purple represents solid particles and yellow represents pores.

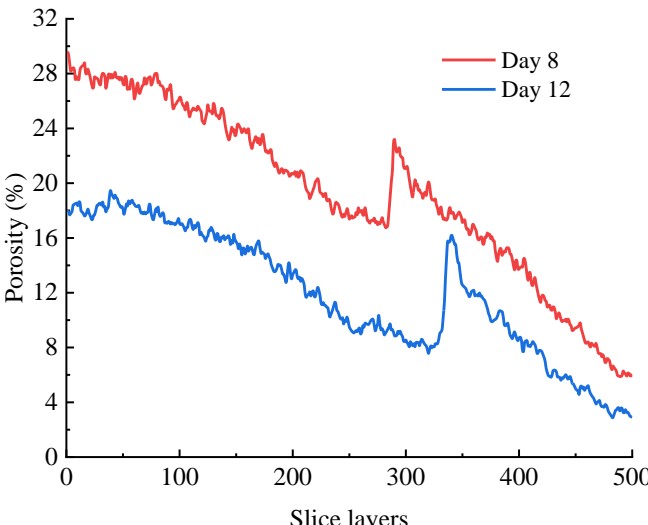

**Figure 11.** Porosity statistics of tailings layer-by-layer at each grouting time.

According to the statistical analysis shown in Figure 11, The variation interval of layer-by-layer porosity for 8 days of grouting ranged from 4.94% to 29.59%, with an average value of 18.85%. Similarly, after 12 days of grouting, the porosity ranged from 2.51% to 19.477%, with an average value of 11.75%. It can be observed from Figures 10 and 11 that the porosity of the tailing column within the same region increases and then decreases with grouting time, indicating that microbial grouting reinforcement significantly affects porosity. Furthermore, the porosity gradually decreases from the bottom to the top of the tailing column with the increase in the number of Z-axis slices, mainly due to slurry being injected from the upper part of the column and penetrating downward layer-by-layer, resulting in the higher accumulation of calcium carbonate precipitates in the upper part than in the lower part. This is consistent with the previous results shown in Figure 10.

### 3.2.5. Determination of Changes in Uranium Concentration

To investigate the effect of microorganisms on the immobilization of radioactive uranium in uranium tailings during microbial mineralization, the discharge solution was taken out to measure the uranium concentration every two days. The results are shown in Figure 12.

According to Figure 12, the uranium concentration in the discharge fluid showed a trend of increasing and then decreasing with increasing grouting time. In contrast, the uranium concentration in the discharge fluid rapidly increased and then gradually stabilized for the blank group, which was only injected with pure water. The uranium concentration in the microbial grouting group was slightly lower than that in the blank group: the maximum uranium concentration was 2.405 mg/L. In contrast, the maximum uranium concentration of the microbial grouting group was 2.021 mg/L. Furthermore, it can also be seen from the figure that the uranium concentration in the discharge fluid reached the highest value at 6 days and then started to significantly decrease. Compared to the blank group, a peak uranium concentration was observed in the discharge fluid on the 8th day and remained high. The uranium concentration in the discharge fluid was 0.533 mg/L after 12 days of grouting, while for the blank group, it was 2.308 mg/L, which is 76.91% less than that of the blank group. Therefore, microbial grouting reinforcement of uranium tailings can effectively lock a portion of the uranium, thereby reducing the uranium concentration in the discharge fluid.

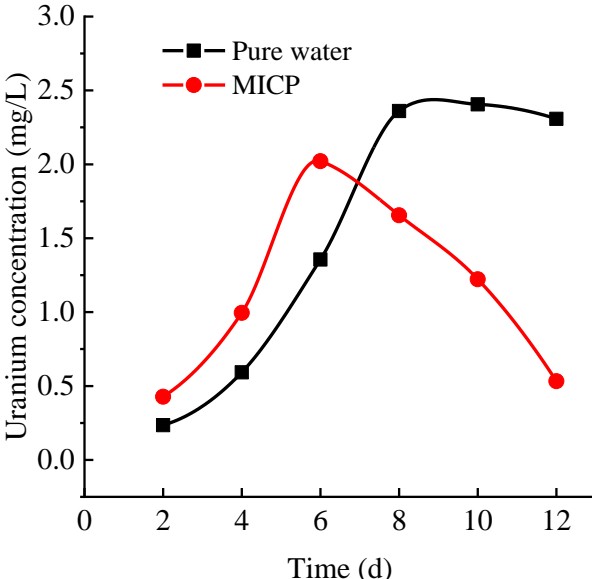

**Figure 12.** Change curve of uranium concentration.

The reason for the change in uranium concentration in the microbial grouting group was analyzed as follows. At the early stage of grouting, the bacterial population is small, resulting in relatively minimal adsorption and fixation of uranium. Meanwhile, the slurry absorbs the uranium in uranium tailing sand. The higher the uranium concentration in the discharge fluid, the longer the soaking time, so the concentration curve shows an upward trend at this stage. With the increased number of grouting cycles, microorganisms multiply rapidly and generate calcium carbonate precipitates that continuously accumulate, and some uranium is adsorbed and fixed in calcium carbonate precipitates. The calcium carbonate also fills in the tailing sand pores, decreasing porosity and hindering the discharge of liquid containing uranium. As a result, the uranium that can be flushed out by the slurry ultimately decreases, and the curve also shows a decreasing trend.

## 4. Discussion

This study demonstrates that MICP has a reinforcing effect on uranium tailings, but further investigations are required to control the uranium content in the tailings. The experiments were conducted under environmental conditions with a pH of 5, which may not apply to tailing ponds with extremely acidic characteristics or tailings located deep within the pond. Therefore, the suitability of MICP-reinforced uranium tailings was studied, mainly due to the significantly reduced microbial activity in highly acidic environments. Furthermore, the uranium concentration in uranium tailings decreased, which may be due to the uranium co-precipitated with calcium carbonate in solid crystals or locked in the tailings particles. The uranium was removed via microbial adsorption, and thus further research of the mechanisms of uranium removal and immobilization is required.

## 5. Conclusions

In this study, solution tests and uranium tailing grouting tests were carried out to systematically investigate the reinforcing effect of MICP on uranium tailing and the control of radionuclides. Based on the results, the following conclusions can be drawn.

The *Sporosarcina pasteurii* grew rapidly, and the bacterial number showed a vigorous linear increase and reproduction under environmental conditions with a pH of 5 and uranium concentration of 2 mg/L. After cementing the solution, the mineralization reactions could be carried out as usual, inducing calcium carbonate precipitation.

The uranium concentration in the solution was effectively reduced during microbially induced calcium carbonate precipitation. The uranium concentration was reduced most

rapidly within the first 10 h, and the efficiency of the removal of elemental uranium was 60.9% at 24 h.

The mechanical strength of uranium tailing sand was improved via microbial grouting for 12 days. Compared with the control group, the cohesion of solidified tailing sand increased by 2.937 times, and the internal friction angle increased by 8.393°. Furthermore, the curve characteristics of stress–strain and ordinary consolidated clay curves were alike, and the peak value of stress increased by 1.87 times at the surrounding pressure of 50 kPa.

The CT scan results show that the tailing sand pores are significantly affected by microbial grouting reinforcement. With a higher number of grouting days, the porosity gradually decreased, and the upper porosity was smaller than the lower.

The uranium concentration of the discharge fluid showed a trend of increasing and then decreasing with the higher number of grouting days. After 12 days, compared to the control group, the uranium discharge concentration decreased by 76.91%.

**Author Contributions:** Conceptualization, L.H. and H.Z.; Data curation, Y.T.; Funding acquisition, Z.Z. and L.W.; Investigation, Y.T.; Methodology, L.W.; Project administration, L.W.; Software, G.H.; Supervision, Z.Z.; Validation, Z.Z. and Q.Y.; Visualization, Q.Y. and G.H.; Writing—original draft, L.H. and H.Z.; Writing—review and editing, L.H. All authors have read and agreed to the published version of the manuscript.

**Funding:** This research was funded by the Research Foundation of Education Bureau of Hunan Province (grant numbers: 22B0410, 22A0291, 20B496), the National Natural Science Foundation of China (grant numbers: 52274167, 52274127), the Natural Science Foundation of Hunan Province (grant numbers: 2022JJ40374, 2023JJ30516, 2021JJ30580), the Hengyang City Science and Technology Program Project Funding (grant numbers: 202150063769), and the Hunan Province's technology research project "Revealing the List and Taking Command" (grant numbers: 2021SK1050).

**Informed Consent Statement:** Not applicable.

**Data Availability Statement:** The data presented in this study are available on request from the corresponding author.

**Conflicts of Interest:** The authors declare no conflict of interest.

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
