# Peer review of "Experimental Study on the Solidification of Uranium Tailings and Uranium Removal Based on MICP"

_sustainability, doi:10.3390/su151612387_

Round 1
Reviewer 1 Report
In the study solution tests and uranium tailing grouting tests were carried out to investigate the reinforcing effect of MICP on uranium tailing and the control of radionuclides. Based on the results, the following conclusions could be drawn. The uranium concentration in the solution was effectively reduced during microbially induced calcium carbonate precipitation. The uranium concentration was reduced rapidly within the first 10 hours, and the efficiency of removal of elemental uranium was 60.9 % at 24 hours. The mechanical strength of uranium tailing sand was improved by microbial grouting for 12 days. Compared with the control group, the cohesion of solidified tailing sand and internal friction angle increased. Furthermore, the tailing sand pores are significantly affected by microbial grouting reinforcement. With the increase of grouting day, the porosity gradually decreases , and the upper porosity was smaller than the lower. The uranium concentration of the discharge fluid showed a trend of increase and then decrease with the increase of grouting day. After 12 days, compared to the control group, the uranium discharge concentration decreased by 76.91%.
This is very valuable work, but it should be published after minor revisions as below.
General Remarks - no
Detailed remarks
1. Point 2.1.3 – some references should be added
2. Point 2.2.2 – some characteristic of uranium tailings should be added
3. Point 2.2.3 – some references should be added.
4. Line 179 – …optical density… should be added before …OD600…
5. Line 233 – some references should be added
6. Figure 3,8,10 should be self-explaining. Please add proper information.
Reviewer 2 Report
Very good article and very well written. The content of the article concerns the management of uranium waste and the improvement of stability, leachability and limiting the release of radionuclide uranium. For this purpose, a uranium removal solution test and a uranium tailings injection test using the microbial calcium carbonate precipitation (MICP) technology were carried out, and the effect of this on the strengthening of uranium tailings and the synchronous control of uranium radionuclide content in it were discussed.
The strengths of the article are: a very good summary, very well presented technology and
Comments:
1. The abbreviation MICP cannot be in the title without explanation. I suggest using other words
2. Please make a list of abbreviations used in the article at the beginning of the article
3. Please highlight the novelty of the research and justify why the article should be addressed to Sustainability. It won't be difficult with such a good analysis of the literature
4. Please check all graphs and description of the axes, e.g. fig 3
The English language is at a high level.
Reviewer 3 Report
The manuscript deals with the MICP biotechnology applied for uranium removal from tailings. The topic is interesting, taking into consideration that MICP has multiple advantages of simultaneous removal of multiple pollutants, environmental protection, and ecological sustainability.
The following comments should improve the scientific quality of the manuscript:
Pag. 3, lines 111-112: the concentration of uranyl ions and the uranium concentration. How the authors made the difference between them?
Line 115: how the concentration of uranium was set at 2 mg/l in the solution experiment?
Please mention the instrumentation used for tests and analyses: pH-meter, OD600 analyzer and the instrumentation used for uranium analysis (name and country of origin). Also, please mention the operation parameters for U analysis instrument. As well for XRD and SEM instruments.
QA/QC is missing in the manuscript.
In Fig. 3 the bacteria mentioned is Bacillus thuringiensis, but previously in the text the Sporosarcina pasteurii was mentioned. Please explain.
Page 7, line 222: “SEM tests were carried out on calcium carbonate…”
In my opinion, it would be useful to merge chapters 3, 4 and 5 in a chapter named “3. Results and discussions”, subdivided in two sub-chapters: 3.1 Solution test results and 3.2 Uranium tailing sand grouting test results.
Round 2
Reviewer 3 Report
The recommendations I made have been taken into considerations. I recommend publishing the article in this new form.